# Mice lacking nucleotide sugar transporter SLC35A3 exhibit lethal chondrodysplasia with vertebral anomalies and impaired glycosaminoglycan biosynthesis

**Soichiro Saito[1]⊙, Shuji Mizumoto[2]⊙, Tsukasa Yonekura[1], Rina Yamashita[1], Kenta Nakano[3], Tadashi Okubo[4], Shuhei Yamada[2], Tadashi Okamura[3], Tatsuya Furuichi[1,5]***

1 Laboratory of Laboratory Animal Science and Medicine, Co-Department of Veterinary Medicine, Faculty of Agriculture, Iwate University, Morioka, Iwate, Japan, 2 Department of Pathobiochemistry, Faculty of Pharmacy, Meijo University, Nagoya, Aichi, Japan, 3 Department of Laboratory Animal Medicine, Research Institute, National Center for Global Health and Medicine (NCGM), Shinjuku-ku, Tokyo, Japan, 4 Department of Laboratory Animal Science, Kitasato University School of Medicine, Sagamihara, Kanagawa, Japan, 5 Laboratory of Laboratory Animal Science and Medicine, Graduate School of Veterinary Sciences, Iwate University, Morioka, Iwate, Japan

⊙ These authors contributed equally to this work.
* furuichi@iwate-u.ac.jp

**Data Availability Statement:** All relevant data are within the paper and its Supporting Information files.

## Abstract

SLC35A3 is considered an uridine diphosphate *N*-acetylglucosamine (UDP-GlcNAc) transporter in mammals and regulates the branching of *N*-glycans. A missense mutation in *SLC35A3* causes complex vertebral malformation (CVM) in cattle. However, the biological functions of SLC35A3 have not been fully clarified. To address these issues, we have established *Slc35a3*$^{-/-}$ mice using CRISPR/Cas9 genome editing system. The generated mutant mice were perinatal lethal and exhibited chondrodysplasia recapitulating CVM-like vertebral anomalies. During embryogenesis, *Slc35a3* mRNA was expressed in the presomitic mesoderm of wild-type mice, suggesting that SLC35A3 transports UDP-GlcNAc used for the sugar modification that is essential for somite formation. In the growth plate cartilage of *Slc35a3*$^{-/-}$ embryos, extracellular space was drastically reduced, and many flat proliferative chondrocytes were reshaped. Proliferation, apoptosis and differentiation were not affected in the chondrocytes of *Slc35a3*$^{-/-}$ mice, suggesting that the chondrodysplasia phenotypes were mainly caused by the abnormal extracellular matrix quality. Because these histological abnormalities were similar to those observed in several mutant mice accompanying the impaired glycosaminoglycan (GAG) biosynthesis, GAG levels were measured in the spine and limbs of *Slc35a3*$^{-/-}$ mice using disaccharide composition analysis. Compared with control mice, the amounts of heparan sulfate, keratan sulfate, and chondroitin sulfate/dermatan sulfate, were significantly decreased in *Slc35a3*$^{-/-}$ mice. These findings suggest that SLC35A3 regulates GAG biosynthesis and the chondrodysplasia phenotypes were partially caused by the decreased GAG synthesis. Hence, *Slc35a3*$^{-/-}$ mice would be a useful model for investigating the *in vivo* roles of SLC35A3 and the pathological mechanisms of SLC35A3-associated diseases.

**Funding:** This study was supported by Grant-in-Aid from the Ito Foundation (T.F.). This study is also supported in part by a Grant-in-Aid for Science Research (C) from the Japan Society for the Promotion of Science (JSPS) (21K0593219 to T.F.; 19K07054 to S.M.; 21K065520 to S.Y.), Grant-in Aid from the Research Center for Pathogenesis of Intractable Diseases from the Research Institute of Meijo University (S.M. and S.Y.), and Grants-in-Aid for Research from the National Center for Global Health and Medicine (20A1019 and 20A2007D to T.O.(Okamura); 21A1018 to K.N.).The funders had no role in study design, data collection and analysis, decision to publish, or preparation of the manuscript.

**Competing interests:** The authors have declared that no competing interests exist.

## Introduction

Most glycosylation reactions take place in the lumen of the endoplasmic reticulum (ER) and the Golgi apparatus [1, 2], in which glycosyltransferases utilize nucleotide sugars as donor substrates and catalyze the transfer of sugar moieties from a nucleotide sugar to a nucleophilic glycosyl acceptor molecule. Most nucleotide sugars are synthesized in the cytosol and cannot cross the lipid bilayer membranes of the ER and Golgi apparatus. Thus, to cross these membranes, they must be transported by a group of nucleotide sugar transporters (NSTs) [3–5], which are highly conserved type III transmembrane proteins with 6–10 transmembrane domains. NSTs transport nucleotide sugars coupled with the antiport of nucleoside monophosphate, a reaction product of luminal nucleoside diphosphatase, which acts on nucleoside diphosphates that are produced in a glycosyltransferase reaction. Several NSTs also transport an activated sulfate, $3'$-phosphoadenosine $5'$-phosphosulfate, which is the universal sulfur donor for all sulfation reactions.

NSTs belong to the solute carrier 35 (SLC35) family and their function is highly conserved among simple eukaryotes, fungi, parasites, plants, and mammals [3–5]. They are classified into seven subfamilies: SLC35A–G. To date, 31 SLC35 genes have been identified in the human genome, and many human and animal genetic studies have identified the NSTs involved in disease development [6, 7]. SLC35A3 is an uridine diphosphate $N$-acetylglucosamine (UDP-$N$-GlcNAc) transporter located in the Golgi membrane [8, 9]. The first reported genetic disorder caused by *SLC35A3* mutations was complex vertebral malformation (CVM), an autosomal recessive severe skeletal dysplasia found in Holstein cattle [10]. CVM is characterized by a range of complex anomalies in the vertebral column and limbs, dysmorphic craniofacial features, cardiac anomalies, and spontaneous abortion or perinatal death. A missense mutation (p.Val180Phe) in bovine *SLC35A3* causes CVM. In humans, compound heterozygous mutations in *SLC35A3* were identified in patients from three unrelated families with arthrogryposis, mental retardation, and seizures (AMRS; MIM 615553) [11–13]. Furthermore, a homozygous missense mutation in *SLC35A3* was identified in a patient with severe vertebral anomalies [14]. Conversely, SLC35A3 has also been reported to regulate the branching of $N$-glycans in cultured cells [15–17]. Decreased levels of highly branched $N$-glycan were also observed in fibroblasts isolated from a patients with AMRS [13]. To date, the role played by SLC35A3 in the synthesis of sugar chains other than $N$-glycans has barely been clarified; therefore, there is a poor understanding of the pathological mechanisms underlying SLC35A3-associated diseases.

Most vertebrate skeletons are formed by a process of endochondral ossification, in which each skeletal element is established and patterned as cartilage that is later replaced by bone [18, 19]. Limb cartilage derives from lateral plate mesoderm, whereas that of the vertebrae and ribs derives from the sclerotomal compartment, i.e., the somites [20, 21]. Somites are segmented blocks of mesoderm formed periodically from the presomitic mesoderm (PSM), and optimal Notch signaling is essential for their formation [22, 23]. Severe vertebral anomalies are the hallmarks of various Notch-associated diseases as well as CVM, and the addition of GlcNAc to the $O$-fucose residue on Notch receptors is essential for its optimal signaling transduction [24, 25]. Collectively, these findings suggest that SLC35A3 transports the UDP-GlcNAc used for the $O$-glycan modification of Notch receptors.

Major components of the extracellular matrix (ECM) of cartilage produced by chondrocytes are type II collagen and aggrecan, the latter of which is a large proteoglycan with numerous chondroitin sulfate (CS) and keratan sulfate (KS) chains attached to its core protein [26–28]. In cartilage tissues, aggrecan contributes to water retention and provides resistance to compression. CS and KS belong to a class of linear sulfated glycosaminoglycan (GAG)

polysaccharides, which are classified into CS, dermatan sulfate (DS), KS, heparan sulfate (HS), and hyaluronan (HA), based on their structural units [29, 30]. CS is composed of repeating disaccharide units of glucuronic acid (GlcUA) and *N*-acetylgalacosamine (GalNAc), and DS is obtained by epimerization of GlcUA in CS chains into iduronic acid (IdoUA), leading to the formation of repeating units of IdoUA and GalNAc. HS and KS consist of repeating disaccharide units of GlcUA/IdoUA and GlcNAc, and of galactose and GlcNAc, respectively. HA is a free-polysaccharide and composed of repeating units of GlcUA and GlcNAc. Cartilage tissue contains not only CS and KS, but also HS, DS and HA. These GAGs are known to play multiple roles in skeletal development and homeostasis [31, 32]. Although the silencing of *SLC35A3* expression in cultured cells has been reported to decrease KS levels but not HS levels [15], the relationship between the function of SLC35A3 and GAG biosynthesis is not fully clarified.

In the present study, we generated a *Slc35a3* knockout (*Slc35a3*$^{-/-}$) mouse strain using the CRISPR/Cas9-mediated genome editing system with the aim of determining the biological functions of SLC35A3 and the pathogenic mechanisms of SLC35A3-related diseases. The generated mutant mice were perinatal lethal and exhibited dwarfism recapitulating CVM-like vertebral anomalies.

## Materials and methods

### Mice and ethical statement

Mice were housed in a temperature-controlled room under a 12/12-h light/dark cycle. They were fed with standard mouse laboratory chow and had free access to water. Mice were sacrificed by an overdose of pentobarbital or cervical dislocation under isoflurane anesthesia. This study was conducted in strict accordance with the guidelines for the Proper Conduct of Animal Experiments (Science Council of Japan). Mouse embryos were sacrificed by decapitation. All animal experiments were approved by the Animal Experimentation Committee at Iwate University (approval no. A202040), the National Center for Global Health and Medicine (approval no. 18037), and Kitasato University School of Medicine (approval no. 2021–130). This study was conducted in strict accordance with the guidelines for the Proper Conduct of Animal Experiments (Science Council of Japan). All efforts were made to minimize animal suffering.

### Genome editing

CRISPR/Cas9-mediated genome editing was performed in mice as described previously but with some modifications [33]. Briefly, crRNA for the target sequence (5′-CTGACGGGATAG CGAGCTTC-3′) and tracrRNA were synthesized by Integrated DNA Technologies (Coralville, IA, USA), and recombinant Cas9 protein was purchased from Integrated DNA Technologies. The gRNA (25 ng/μL; a complex of crRNA and tracrRNA) and Cas9 protein (50 ng/μL) were preincubated at 37°C for 15 min, and then the two were coelectroporated into the cytoplasm of fertilized eggs derived from C57BL/6J mice (Japan SLC, Hamamatsu, Japan). The super-electroporator NEPA21 (NEPA GENE, Chiba, Japan) and Petri dish platinum plate electrodes (length: 10 mm; width: 3 mm; height: 15 mm; gap: 5 mm; NEPA GENE) were used. The electric condition of the poring pulses was fixed as follows: voltage, 150 V; pulse width, 1.0 ms; pulse interval, 50 ms; number of pulses, +4 [34]. The electric condition of the transfer pulses was set as follows: voltage, 20 V; pulse width, 50 ms; pulse interval, 50 ms; number of pulses, ±5. After the electroporated oocytes were cultured overnight in vitro, two-cell embryos were transferred into pseudopregnant female mice. Genomic DNA was isolated from tail samples of the offspring using standard methods. The edited region around exon 3 of the *Slc35a3* gene was amplified using PCR with Ex Taq (TaKaRa Bio, Otsu, Japan) and the following primers:

5′- `ACTGATTTTCCCCATCGTGT`-3′ and 5′-`CTTGCTTGAGGTAATGG`-3′. The amplification products were sequenced and their sequences were compared with the reference sequence.

## Genotyping of the 17-pb deletion allele in *Slc35a3*

A stable germline transmission of the edited 17-bp deletion allele in *Slc35a3* was confirmed, and genotyping of the mouse line lacking the 17-bp was performed using agarose gel electrophoresis of the PCR products amplified with Ex Taq and the following primers: 5′-`TGTGAGAGCACTGAATAGAGT`-3′ and 5′-`TGGCTGCATCTAGGTTTGAC`-3′.

## Real-time PCR

Total RNA was extracted from the ribs at embryonic day (E) 18.5 using RNAiso Plus (Takara Bio). Equal amounts of total RNA were reverse-transcribed into cDNA using ReverTra Ace qPCR RT Master Mix (TOYOBO, Tokyo, Japan). Each reverse transcription reaction (1 μl) was used as a template for real-time PCR using THUNDERBIRD SYBR qPCR Mix (TOYOBO). The following primers were used for amplification: 5′-`ACCTAGATGCAGCCACTTAC`-3′ and 5′-`AGCCACTGGTACACACCTAA`-3′ for *Slc35a3* and 5′-`ATGTGTCCGTCGTGGATCTG`-3′ and 5′-`AGGTGGAAGAGTGGGAGTTG`-3′ for *Gapdh*. SYBR Green PCR and real-time fluorescence detection were performed using StepOnePlus™ Real-Time PCR System (Thermo Fisher Scientific, MA, USA). The expression levels of *Slc35a3* mRNA were normalized to those of *Gapdh* mRNA.

## Skeletal specimen

Sacrificed mice were eviscerated and fixed in 99% EtOH for 4 days. Alcian blue staining was performed using a solution containing 80% EtOH, 20% acetic acid, and 0.015% Alcian blue for 4 days at 37°C. Specimens were then rinsed and soaked in 95% EtOH for 3 days. Subsequently, Alizarin red staining was then performed using a solution containing 0.002% Alizarin red and 1% KOH for 12 h at room temperature. After the specimens were rinsed with water, they were kept in a 1% KOH solution until the skeletons became clearly visible. For storage, specimens were sequentially transferred into 50%, 80%, and 100% glycerol.

## Histological analyses

Limbs and lumbar vertebrae dissected from sacrificed embryos were fixed in 4% PFA for 24 h at 4°C, decalcified in 10% (w/v) EDTA /7% (v/v) glycerol (pH 7.4) for 48 h at 4°C, and embedded in paraffin. Hematoxylin and eosin staining was performed using 6-μm paraffin sections according to standard protocols. Apoptotic cells were detected using TUNEL staining with an *In Situ* Apoptosis Detection Kit (TaKaRa). The frequencies of BrdU- and TUNEL-positive cells were calculated using ImageJ software. *In situ* hybridization was performed by Genostaff Co., Ltd (Tokyo, Japan) as described previously [35]. For the analysis of cell proliferation, we injected pregnant mice intraperitoneally with 100 μg of BrdU/g of body weight 1 h before sacrifice. BrdU-labeled cells incorporated on the sections from BrdU-treated embryos were stained with a BrdU Immunohistochemistry Kit (Abcam, Cambridge, MA, UK).

## Whole-mount *in situ* hybridization

ICR mouse embryos delivered by cesarean section at E9.25 were fixed in 4% PFA/PBS overnight at 4°C and dehydrated. The pBluescript II vector used for inserting mouse *Slc35a3* cDNA (c.406–919) was linearized. The antisense or sense probes were then synthesized using

DIG-RNA labeling mix with T7 or T3 RNA polymerases (Roche Life Science, Penzberg, Germany). To perform whole-mount *in situ* hybridization, the embryos were rehydrated in PBS containing 0.1% Tween 20 (PBST) and then treated with 20 μg/mL proteinase K in PBST for 7–8 min. After refixation in 4% PFA/0.2% glutaraldehyde (Sigma Aldrich, St. Louis, MO, USA), the embryos were hybridized overnight at 66˚C with DIG-labeled RNA probes. After posthybridization washing, the embryos were incubated with the anti-DIG antibody coupled with alkaline phosphatase (Roche Life Science) diluted 1:2000 in Tris-buffered saline containing 0.1% Tween 20 (TBST) for 3 h at room temperature. After TBST washing, alkaline phosphatase activity was visualized using BM-Purple (Roche Life Science).

## Disaccharide composition analysis of glycosaminoglycans

The amounts of total disaccharides of CS/DS, HS, KS, and HA in the spine and limbs of mouse embryos at E18.5 were determined as described previously [29]. Briefly, each tissue was homogenized, sonicated, and treated exhaustively with actinase E (Kaken Pharm. Kyoto, Japan) to degrade proteins. Total amount of proteins in the sonicates was determined by micro BCA protein assay kit (Thermo Fisher Scientific). The remaining proteins and peptides were precipitated using trichloroacetic acid, and the supernatant was followed by extraction with ether to remove trichloroacetic acid. The resultant crude GAG-peptide fractions were desalted using an Amicon Ultra-4 3 K unit (Millipore, Billerica, MA, USA), and treated individually with a mixture of chondroitinase ABC and chondroitinase AC-II (Seikagaku Corp., Tokyo, Japan), a mixture of heparinase-I, heparinase-III (IBEX Pharmaceuticals, Montreal, Canada), and heparitinase-II (R&D Systems, Minneapolis, MN, USA), or keratanase-II (Seikagaku Corp.) for analysis of the disaccharide composition of CS/DS/HA, HS or KS, respectively. The digests were labeled with a fluorophore 2-aminobenzamide (2AB), and aliquots of the 2AB-derivatives of GAG disaccharides were analyzed using anion-exchange HPLC on a PA-G column (YMC Co., Kyoto, Japan). Unsaturated disaccharides detected in the digests were identified via comparisons with the elution positions of authentic 2AB-labeled disaccharide standards.

## Statistical analyses

Student's *t*-tests were used to determine the significance of differences between the wild-type (WT) and $Slc35a3^{-/-}$ or the Control (Ctrl, WT + $Slc35a3^{+/-}$) and $Slc35a3^{-/-}$ mouse groups. The significance level was defined as $P < 0.05$.

## Results

### Generation of $Slc35a3^{-/-}$ mice by genome editing

To generate $Slc35a3^{-/-}$ mice, we used the CRISPR/Cas9 system to target exon 3 of *Slc35a3*. In total, 360 injected zygotes were transferred to pseudopregnant mice; consequently, 32 F0 pups were born. Among these pups, eight died after birth and exhibited dwarfism. DNA sequencing around the target site showed that five lethal pups had frameshift deletions in both *Slc35a3* alleles. The DNA sequences of the other three lethal pups could not be determined. The 24 F0 pups with normal body size had either one or no targeted allele. Phenotypic differences were not observed between the WT mice and the heterozygotes for the edited allele. These results suggest that the lethal and dwarf phenotypes were caused by SLC35A3 deficiency. A stable germline transmission of the edited allele was confirmed in several heterozygous F0 mice, and the $Slc35a3^{-/-}$ mice used in further experiments were generated by crossing of the heterozygous breeding pairs lacking 17-bp of the coding sequence (c. 255_271del) (Fig 1A). Embryos at

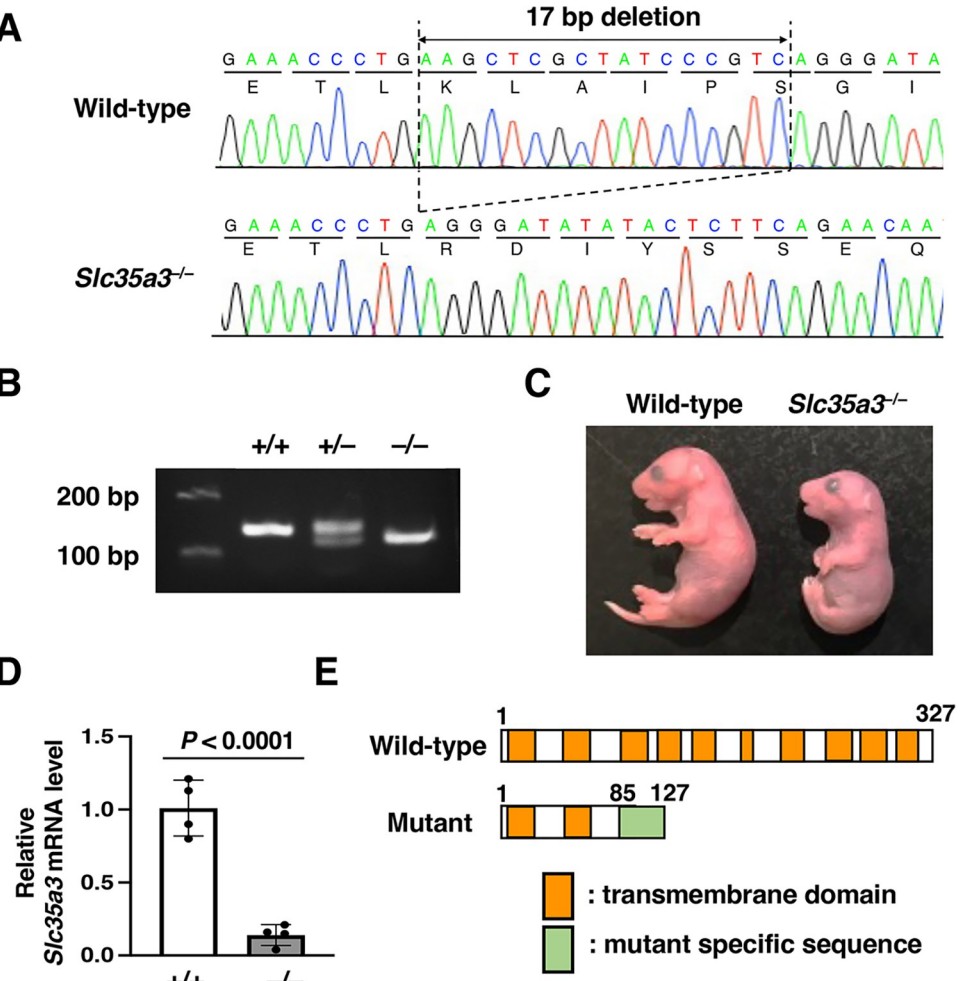

**Fig 1. Generation of *Slc35a3*⁻/⁻ mice using genome editing.** (A) Sequence chromatograms of the edited genomic region in exon 3 of the *Slc35a3* gene. A homozygous 17-bp deletion (c.255_271del) leading to frameshift occurred in *Slc35a3*⁻/⁻ mice. (B) Agarose gel electrophoresis image of PCR products used for genotyping. +/+: wild-type; +/−: *Slc35a3*⁺/⁻; −/−: *Slc35a3*⁻/⁻. Upper band: wild-type allele-specific PCR products; lower band: 17-bp deletion allele-specific PCR products. (C) Gross appearance of wild-type and *Slc35a3*⁻/⁻ littermate embryos at E18.5. (D) Relative *Slc35a3* mRNA expression in the limbs of wild-type (+/+) and *Slc35a3*⁻/⁻ mice (−/−). Data are shown as means ± standard deviation (n = 4). *P* values were calculated using Student's *t*-test. (E) Structural comparison of wild-type and mutant SLC35A3 proteins. The wild-type SLC35A3 protein consists of 327 amino acids with 10 predicted transmembrane domains, whereas the mutant protein consists of 85 SLC35A3-specific amino acids with 2 transmembrane domains and 42 mutant specific amino acids.

E18.5 were delivered via cesarean section and genotyped using PCR and agarose gel electrophoresis (Fig 1B). *Slc35a3*⁻/⁻ mice were identified according to the Mendelian ratio and died within an hour of birth while exhibiting severe dwarfism (Fig 1C). The expression levels of *Slc35a3* mRNA in the limbs of the mutants were drastically reduced compared with those of WT mice (Fig 1D). The WT SLC35A3 protein consists of 327 amino acids with 10 predicted transmembrane domains (Fig 1E) [36]. If the mutant proteins are produced form fewer transcripts of the 17-bp deletion allele, they should consist of 85 SLC35A3-specific amino acids with 2 transmembrane domains and 42 mutant specific amino acids (Fig 1E). Based on the activities of the truncated mutant NST proteins reported previously [37], the predicted

SLC35A3 mutant protein should lose almost all of its transport activity. Therefore, the 17 bp-deletion could be a null or a strong loss-of-function mutation.

**Vertebral anomalies are reproduced in *Slc35a3*$^{-/-}$ mice.** To investigate the effect of SLC35A3 deficiency on skeletogenesis, we prepared the skeletal specimens of WT and *Slc35a3*$^{-/-}$ littermate embryos at E18.5. The mutant mice exhibited the shortening of most of the skeletal elements (Fig 2A). The skulls of *Slc35a3*$^{-/-}$ mice appeared to be slightly smaller than those of WT mice (Fig 2B). The long bones, such as femur and tibia, of the mutants were significantly shorter than those of Ctrl mice (Fig 2C and 2D). Delayed fusion of the sternum was observed in the mutants (Fig 2E). *Slc35a3*$^{-/-}$ mice exhibited severe vertebral and rib anomalies, including fusion and malformations of the thoracic and lumbar vertebrae, and a decreased number and bifurcation of ribs (Fig 2F). Individual differences were observed in the number of rib pairs of *Slc35a3*$^{-/-}$ mice (11–13 pairs), whereas WT mice had 13 pairs. Vertebrae and ribs are derived from the somites, which are formed periodically from the PSM at the tail end of the embryo. During mouse embryogenesis, somite formation commences from E8.0 and ends at E13.0. Whole-mount *in situ* hybridization using WT embryos at E9.25 was performed to investigate *Slc35a3* mRNA expression during somite formation. Positive signals were detected in the truncal site of the embryos, and these became stronger as the proximity to the tail end decreased (Fig 3A). In magnified images of the tail region, positive signals were detected in the PSM and the neuroepithelium (Fig 3B).

## Cartilaginous ECM quality is impaired in *Slc35a3*$^{-/-}$ mice

To understand the pathogenic basis for chondrodysplasia in *Slc35a3*$^{-/-}$ mice, we investigated the histology of the cartilaginous tissues in the humerus and lumbar vertebrae at E18.5. Humeral epiphyseal growth plate is composed of three distinct types of chondrocytes, i.e., round, flat, and hypertrophic chondrocytes (Fig 4A and 4C). Round chondrocytes at the end of cartilage are termed as periarticular chondrocytes and form the joint surface. Other round chondrocytes proliferate and produce cartilage ECM composed of collagen fibrils and proteoglycans. Round chondrocytes differentiate into flat chondrocytes, which continue to proliferate and form columns along the longitudinal axis. Flat chondrocytes differentiate into hypertrophic chondrocytes, which undergo apoptosis or transdifferentiate into osteoblasts. The growth plate of *Slc35a3*$^{-/-}$ mice was significantly shorter than that of WT mice (Fig 4B). The round and flat chondrocytes were well spaced by ECM in WT mice (Fig 4Ca and 4Cb). In contrast, the ECM space was severely decreased in *Slc35a3*$^{-/-}$ mice (Fig 4Cd and 4Ce); therefore, cell density was significantly increased in the round chondrocyte areas of these mice (Fig 4D). Most flat chondrocytes were reshaped to a spindle-like chondrocytes in the mutants (compare Fig 4Cb and 4Ce). These abnormalities were also observed in the lumbar vertebrae of the *Slc35a3*$^{-/-}$ mutants (Fig 4E and 4F). A BrdU incorporation study showed that the proliferative activities of both round and flat chondrocytes were comparable between Ctrl and *Slc35a3*$^{-/-}$ mice (Fig 5A and 5B). The frequency of TUNEL-positive apoptotic hypertrophic chondrocytes was also comparable between these mice (Fig 5C and 5D). We also performed *in situ* hybridization analysis to detect chondrocyte marker mRNAs. *Col2a1* and *Col10a1* were utilized as the markers for proliferating and hypertrophic chondrocytes, respectively. Signals of these two markers were detected in the mutants, as well as in the WT mice (Fig 5E). Although the heights of the *Col2a1*- and *Col10a1*-expressing zones in *Slc35a3*$^{-/-}$ mice were significantly lower than those in Ctrl mice, the relative heights of the *Col2a1*- and *Col10a1*-expressing zones normalized by the total heights of the growth plate were comparable between these mouse groups (Fig 5F). Therefore, chondrocyte proliferation, differentiation and apoptosis were not affected in *Slc35a3*$^{-/-}$ mice.

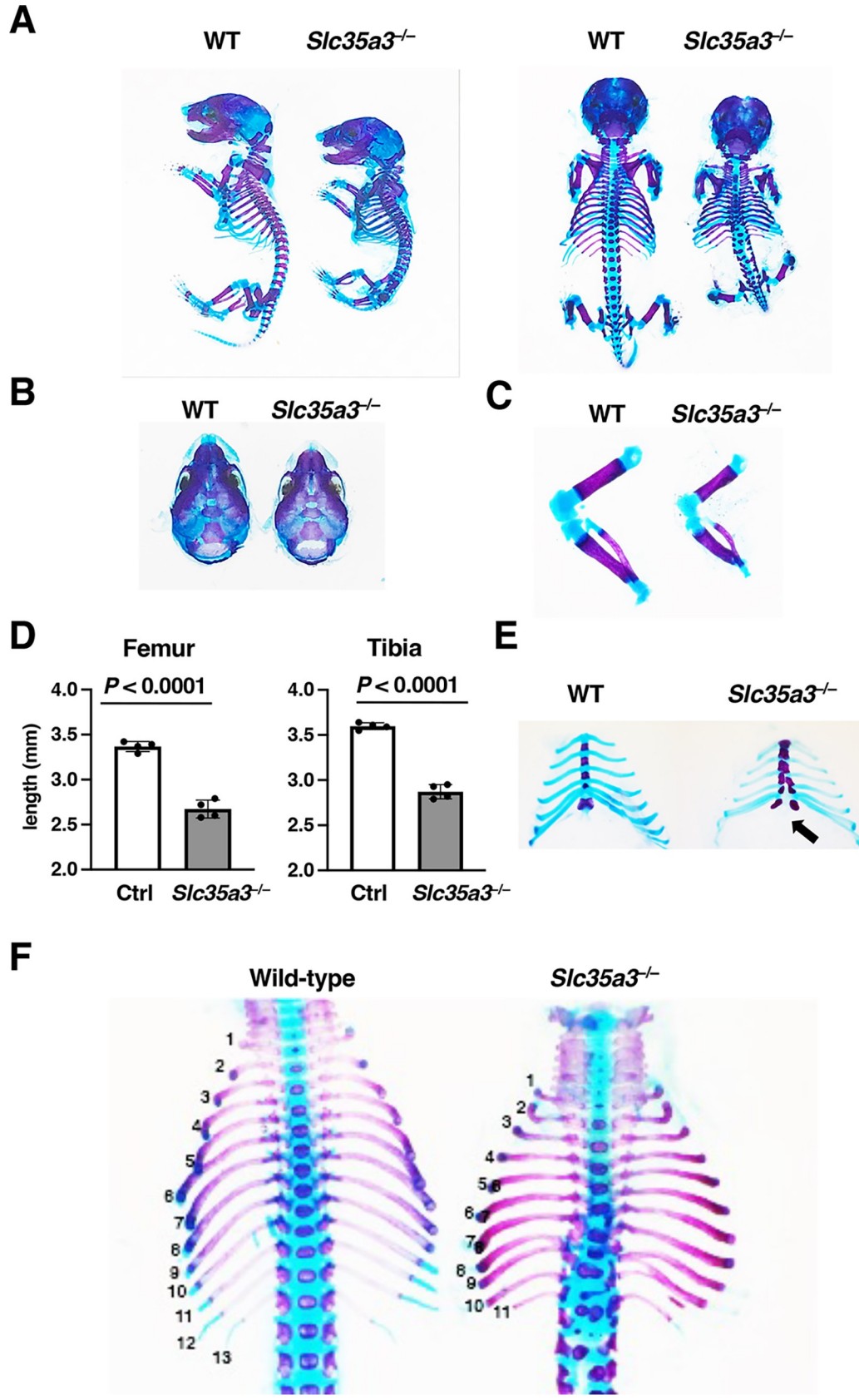

**Fig 2. Skeletal abnormalities in *Slc35a3*⁻/⁻ mice.** Skeletal specimens of control (Ctrl, wild-type + *Slc35a3*⁺/⁻) and *Slc35a3*⁻/⁻ littermate embryos at E18.5 were prepared. Calcified tissues were stained red with Alizarin red, and cartilage was stained blue with Alcian blue. (A) Whole skeleton. (B) Skull. (C) hind limb. (D) Lengths of the femur and tibia in Ctrl and *Slc35a3*⁻/⁻ embryos were measured. Data are shown as means ± standard deviation (n = 4). *P*-values were calculated using Student's *t*-test. (E) Sternum and ribs. The black arrow indicates delayed fusion of the sternum in *Slc35a3*⁻/⁻ mice. (F) Dorsal view of the spine and ribs. *Slc35a3*⁻/⁻ mice exhibited severe vertebral and rib anomalies, including fusion and malformations of the thoracic and lumber vertebrae and a reduced number and bifurcation of ribs. Wild-type mice had 13 rib pairs, whereas *Slc35a3*⁻/⁻ mice had 11 rib pairs. WT, wild-type.

## Glycosaminoglycan biosynthesis is impaired in the spine and limbs of *Slc35a3*⁻/⁻ mice

In the cartilage of *Slc35a3*⁻/⁻ mice, the extracellular space was drastically reduced, and many flat proliferative chondrocytes were reshaped (Fig 4). Because similar histological abnormalities have been observed in several mutant mice accompanied by impaired GAG biosynthesis [37–39], we measured the total amounts and disaccharide compositions of CS/DS, HS, KS, and HA in the spine and limbs of Ctrl and *Slc35a3*⁻/⁻ mice at E18.5, using a combination of enzymatic treatment and anion-exchange HPLC (Fig 6 and S1–S3 Figs; S1–S4 Tables). The amounts of CS/DS and HS-derived disaccharides in the GAG-peptide fractions prepared from the spine and limbs of *Slc35a3*⁻/⁻ mice were significantly decreased compared with those from Ctrl mice (Fig 6A and 6B, S1 and S2 Figs; S1 and S2 Tables). In *Slc35a3*⁻/⁻ mice, the amount of KS in the spine of *Slc35a3*⁻/⁻ mice were also significantly decreased compared with the amounts in Ctrl mice (Fig 6C and S3 Fig; S3 Table). The amounts of HA were comparable between Ctrl and *Slc35a3*⁻/⁻ mice (Fig 6D and S1 Fig; S4 Table).

## Discussion

*Slc35a3*⁻/⁻ mice generated using the CRISPR/Cas genome editing system were perinatal lethal and exhibited dwarfism with severe vertebral anomalies similar to those in CVM. In vertebrates, somites which give rise to vertebrae, ribs, skeletal muscles, and some dermis, are produced sequentially from the PSM at the tail end of the embryo [20, 21]. Somite formation is regulated by the synchronous oscillation of gene expression in the PSM, i.e., the segmentation clock, and Notch signaling is crucial for the "ticking" of this clock [22–25]. Notch receptors

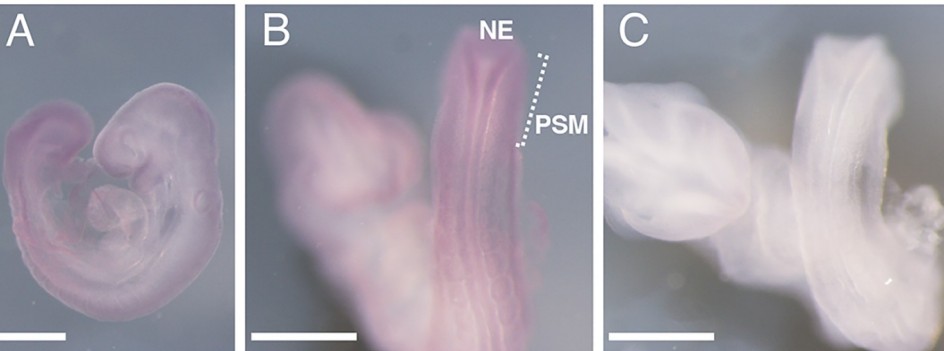

**Fig 3. *Slc35a3* mRNA was expressed in the presomitic mesoderm during somite formation.** Whole-mount *in situ* hybridization images of wild-type mouse embryos at E9.25 (somite 20–22) produced using the antisense (A, B) and the sense (C) probes against *Slc35a3* mRNA. Lateral side view (A), and the magnified images of the tail region (B, C). NE: neuroepithelium, PSM: presomitic mesoderm. Scale bars: 500 μm. The same results were obtained from two independent experiments.

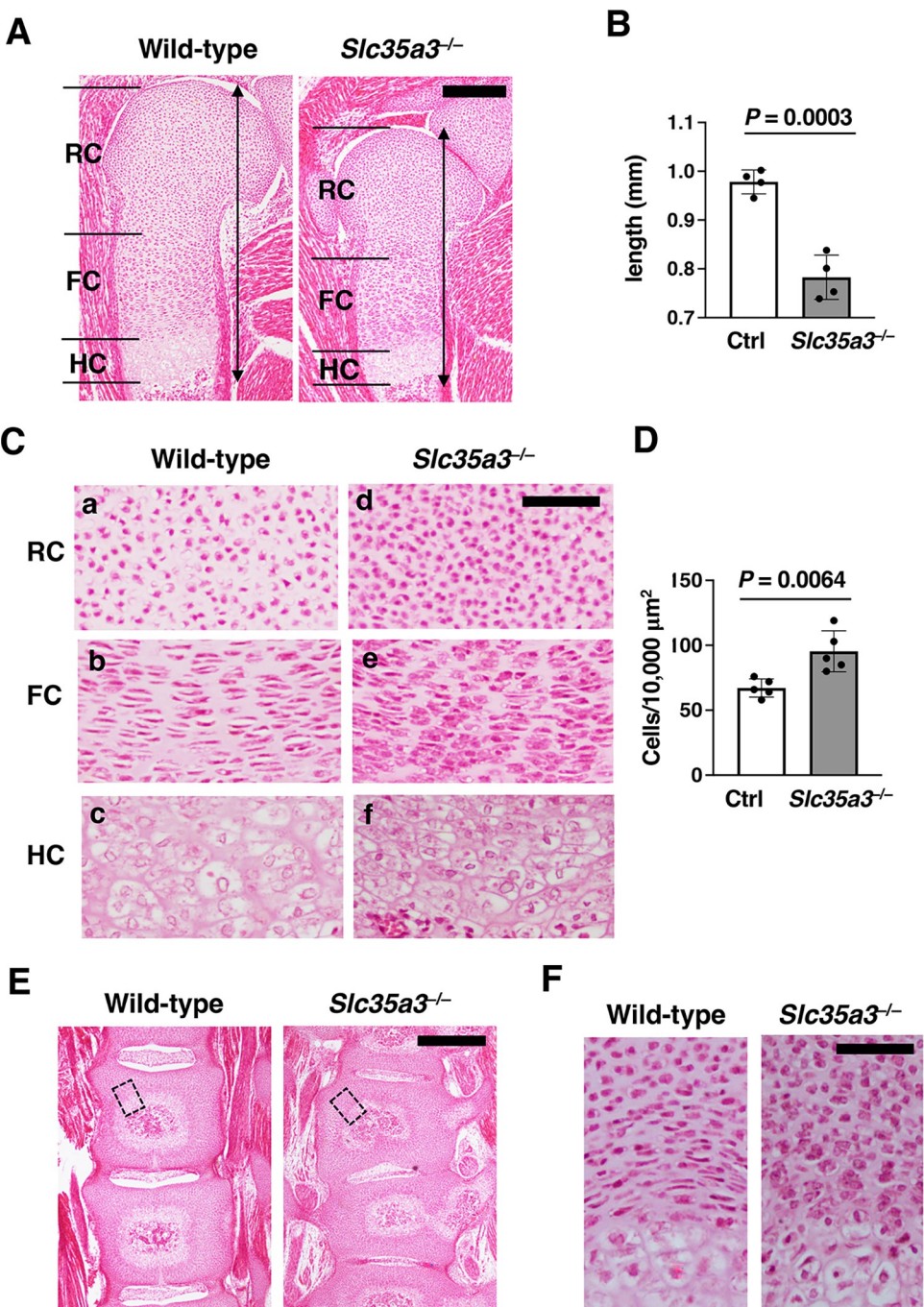

**Fig 4. Cartilaginous extracellular matrix space was severely reduced in *Slc35a3*<sup>−/−</sup> mice.** Humerus and lumbar vertebrae sections of Control (Ctrl, wild-type + *Slc35a3*<sup>+/−</sup>) and *Slc35a3*<sup>−/−</sup> littermate embryos at E18.5 were stained using hematoxylin and eosin (HE) solution. (A) Low magnification images of the humerus. Black double arrows indicate the length of the distal growth plate. (B) Lengths of the humerus growth plate in the HE-stained sections prepared from Ctrl and *Slc35a3*<sup>−/−</sup> embryos were measured. (C) High magnification images of the round chondrocyte (RC), flat chondrocyte (FC), and hypertrophic chondrocyte (HC) zones in the growth plates. (D) Cells were counted in the round chondrocyte zone of HE-stained sections within the squares (100 × 100 μm). (E) Low magnification images of the lumbar vertebrae. (F) High magnification images of the lumbar vertebrae. The boxed regions in (E) are shown at a higher magnification in (F). Data are shown as the means ± standard deviation; n = 4 (B) and 5 (D). *P* values were calculated using Student's *t*-test. Ctrl, control. Scale bars, 200 μm (A, E) and 50 μm (C, F).

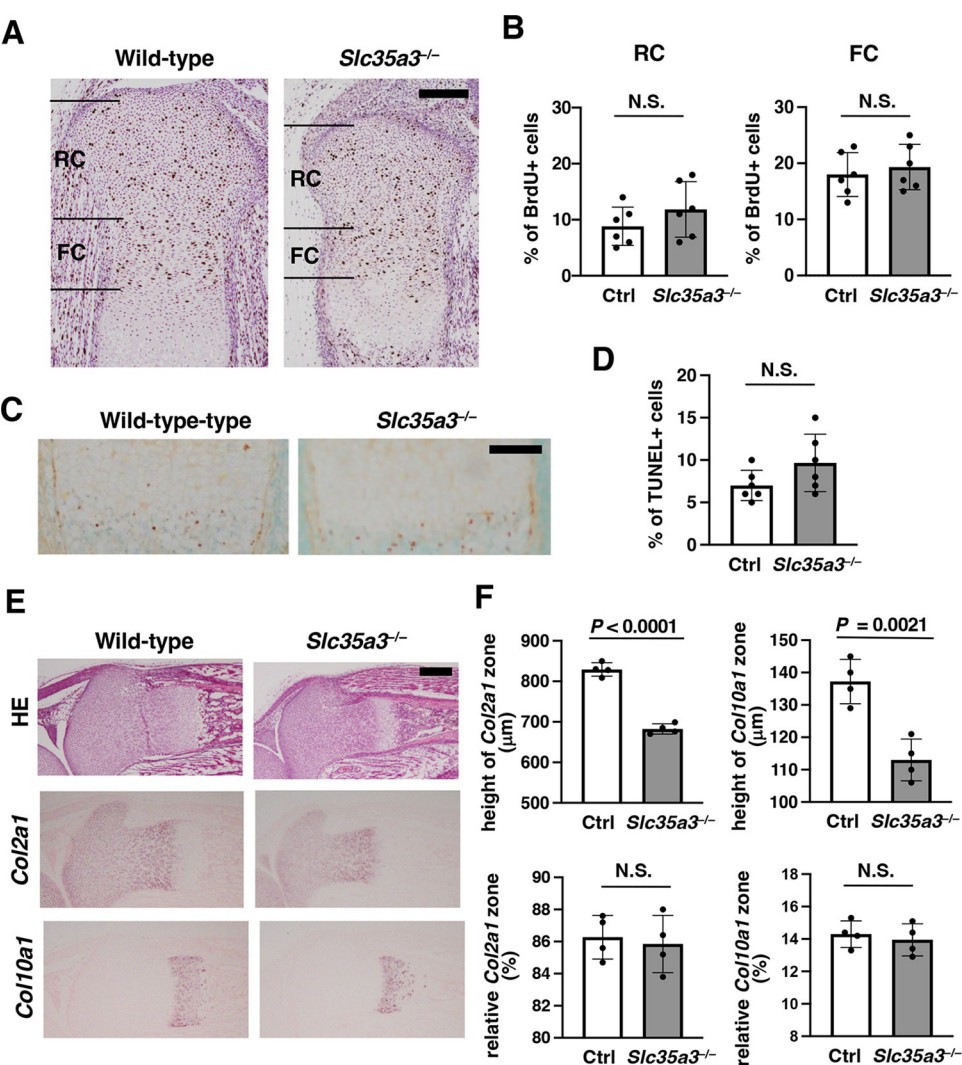

**Fig 5. Proliferation, apoptosis, and differentiation are not affected in *Slc35a3*⁻/⁻ chondrocytes.** (A) BrdU staining of the tibia sections from the BrdU-treated embryos at E18.5. RC, round chondrocyte zone; FC, flat chondrocyte zone. (*B*) Frequency of BrdU-positive cells in the RC and FC zones. (C) TUNEL staining of the hypertrophic chondrocyte zone in the tibia sections at E18.5. (D) Frequency of TUNEL-positive cells in the hypertrophic chondrocyte zone. (E) HE staining and *in situ* hybridization (ISH) images, produced using *Col2a1* and *Col10a1* probes of the tibia sections at E18.5. (F) Heights of *Col2a1*- and *Col10a1*-expressing zones (upper graphs), and the relative heights of *Col2a1*- and *Col10a1*-expressing zones normalized by the total heights of the growth plate (lower graphs) in the ISH images. Data are shown as the means ± standard deviation; n = 6 (B, D) and 4 (F). Combined values from three (B, D) and two (F) independent sections were compared. Two separate areas were counted in one section (B, D, F). *P* values were calculated using Student's *t*-test. Ctrl, control (wild-type + *Slc35a1*⁺/⁻); N.S., not significant. Scale bars, 200 μm (A, E) and 100 μm (C).

are type-I transmembrane proteins with large extracellular domains, containing 29–36 epidermal growth factor-like repeats, most of which are modified with three different types of *O*-linked glycans, i.e., *O*-fucose, *O*-glucose, and *O*-GlcNAc [24, 25]. These unique *O*-glycan modifications are important for optimal Notch activity. LFNG, also known as Lunatic Fringe, is a glycosyltransferase that transfers GlcNAc to this *O*-fucose residue, resulting in the formation of a tetrasaccharide, Sia-Gal-GlcNAc-Fuc, which regulates various steps in Notch pathway activation, including receptor folding, trafficking and ligand interaction [24, 25, 40, 41]. Loss-of-function mutations in *LFNG* cause severe vertebral anomalies in humans and mice [42–44];

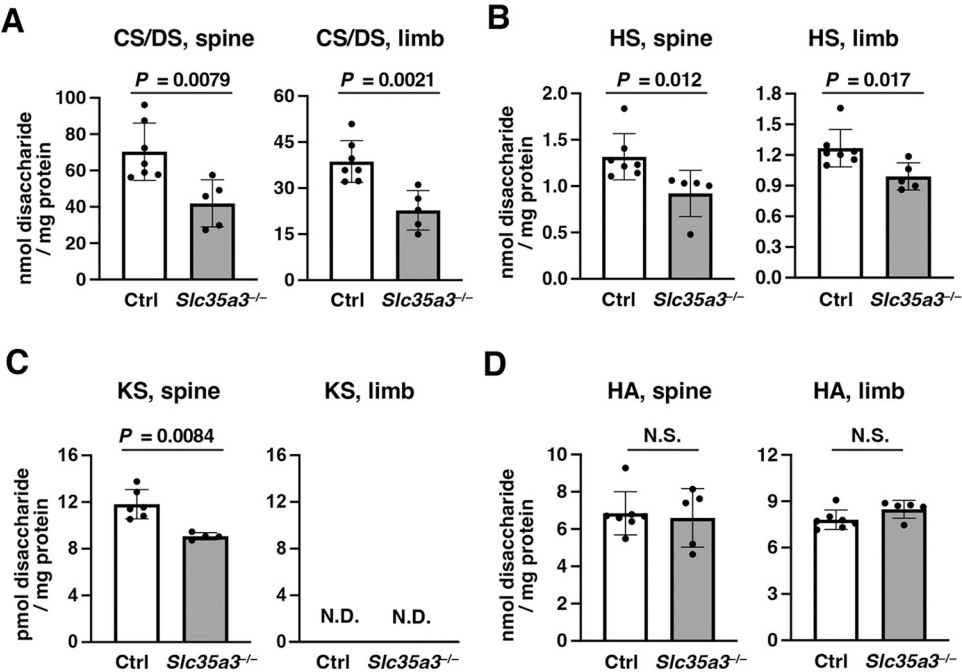

**Fig 6. Glycosaminoglycan (GAG) synthesis is impaired in the spine and limbs of *Slc35a3*<sup>−/−</sup> mice.** The amounts of chondroitin sulfate /dermatan sulfate (CS/DS) (A), heparan sulfate (HS) (B), keratan sulfate (KS) (C), and hyaluronan (HA) (D) in the spine and limbs of control (Ctrl, wild-type + *Slc35a3*<sup>+/−</sup>) and *Slc35a3*<sup>−/−</sup> mice were measured using a combination of enzymatic treatment and anion-exchange HPLC. The GAG fractions prepared from the spine and limbs were individually digested using a mixture of chondroitinases ABC and chondroitinase AC-II, heparinase-I, heparitinase-II, and heparinase-III, or keratanase-II for the specific analyses of CS/DS/HA, HS, or KS, respectively, and each digest was labeled with 2-aminobenzamide (2AB) and analyzed using anion-exchange HPLC (S1–S3 Figs). Amounts of the resultant disaccharides in each sample were calculated based on the peak area in each chromatogram. Data are shown as means ± standard deviation; n = 7 (Ctrl) and 5 (*Slc35a3*<sup>−/−</sup>) in CS/DS, HS, and HA measurements; n = 6 (Ctrl) and 4 (*Slc35a3*<sup>−/−</sup>) in KS measurement. *P* values were calculated using Student's *t*-test. N.S., not significant; N.D., not detected.

therefore, the addition of GlcNAc to the *O*-fucose residue on Notch receptors is essential for its optimal signaling and somite formation. The identification of *SLC35A3* as a CVM causative gene suggests that SLC35A3 is the major NST providing a substrate for LFNG into the Golgi apparatus [10]. In this study, we showed that *Slc35a3*<sup>−/−</sup> mice recapitulated CVM-like vertebral anomalies (Fig 2), and *Slc35a3* mRNA was expressed in the PMS during somite formation (Fig 3), which supports the hypothesis that SLC35A3 is the major NST that provides a substrate for LFNG. To investigate this hypothesis, the expression of Notch signaling molecules involved in the segmentation clock should be examined in detail during somite formation of *Slc35a3*<sup>−/−</sup> embryos.

*Slc35a3*<sup>−/−</sup> mice exhibited severe dwarfism, which was shown to involve a shortened growth plate, decreased ECM space, and deformed flat chondrocytes via histological analyses (Fig 4). Because chondrocyte proliferation, apoptosis and differentiation were not affected in *Slc35a3*<sup>−/−</sup> mice (Fig 5), the chondrodysplasia phenotypes should be caused mainly by abnormalities in ECM quality. In GAG disaccharide composition analyses, the amounts of CS/DS and HS in the spine and limbs and the amount of KS in the spine of *Slc35a3*<sup>−/−</sup> mice were significantly decreased compared with the amounts in Ctrl mice (Fig 6). Similar histological abnormalities have been reported in the two mutants exhibiting lethal chondrodysplasia, *Slc35d1*<sup>−/−</sup> mice and mice lacking both chondroitin sulfate *N*-acetylgalactosaminyltransferase (CSGalNAcT)-1 and -2 [37, 38]. Nucleotide sugar transporter SLC35D1 acts as a general UDP-sugar transporter

located in the ER membrane, and the total GAG level and CS chain length are reduced in the cartilage of $Slc35d1^{-/-}$ mice to approximately a third and a half, respectively, of those in Ctrl mice [37, 45, 46]. CSGalNAcT-1 and -2 are essential for glycosyltransferases in CS biosynthesis, and the CS level in the cartilage of the double knockout mice is reduced to approximately two-fifths of that in the WT mice [38]. In the present study, the CS/DS level in the limbs of $Slc35a3^{-/-}$ mice was reduced to approximately three-fifths of that in the Ctrl mice (Fig 6A), suggesting that the decreased CS level should contribute to the severe chondrodysplasia phenotype of $Slc35a3^{-/-}$ mice. As well as being important in high water retention and elasticity in cartilage tissues [26–28], CS may also be important in securing ECM space and maintaining the morphology of flat chondrocytes. HS has also been reported to have important functions for skeletal development [47, 48]. $Ext1$ encodes a HS glycosyltransferase, and $Ext1^{gt/gt}$ mice possessing the hypomorphic allele of $Ext1$ were found previously to die by E16.5; moreover, the HS level in the mutants was reduced to 20% of that in their Ctrl littermates [49]. Fused vertebrae and disorganized flat chondrocyte were observed in the mutants, as well as $Slc35a3^{-/-}$ mice. It also has been reported that EXT1 is involved in the regulation of bone morphogenetic protein (BMP) signaling [50] and BMP enhanced ECM production in cartilage [51, 52]. Therefore, the decreased GAG synthesis may deteriorate the ECM quality, leading to chondrodysplasia phenotypes in $Slc35a3^{-/-}$ mice.

Because HS and KS contain GlcNAc residues in the repeating disaccharide unit [29, 30], SLC35A3 deficiency should perturb the transport of UDP-GlcNAc, which is used as a donor substrate for HS and KS synthases, into the Golgi lumen, which would lead to decreased HS and KS levels (Fig 6B and 6C). However, the amounts of CS/DS in $Slc35a3^{-/-}$ mice were also significantly decreased (Fig 6A). CS/DS do not contain GlcNAc residues in their units; thus, these defects cannot be explained only by the diminished transport activity of UDP-GlcNAc. SLC35A3 has been reported to function not only as a UDP-GlcNAc transporter, but also as a member of multiprotein complexes with other NSTs and glycosyltransferases, thereby affecting some specific steps in glycosylation [15–17, 53–57]. SLC35A2 and SLC35A3 form a heterologous complex in the Golgi membrane and appear to be functionally coupled [53–57]. SLC35A2 transports UDP-galactose and UDP-GalNAc [3–5], and SLC35A3 partially supports this transport activity. Furthermore, these two NSTs are known to form huge complexes with mannoside acetylglucosaminyltransferases (MGATs) in the Golgi membrane, a process considered to facilitate the biosynthesis of complex-$N$-glycans [15, 16, 54]. Therefore, the functional coupling of SLC35A3 and UDP-GalNAc transporters such as SLC35A2, in $Slc35a3^{-/-}$ cells, may be disrupted, leading to a UDP-GalNAc shortage in the Golgi lumen and decreased CS/DS biosynthesis. It is also plausible that SLC35A3 deficiency affects the catalytic activities of glycosyltransferases involved in CS/DS biosynthesis. To investigate these hypotheses, further investigations including the identification of multiprotein complexes involving SLC35A3 in cartilage should be conducted. Conversely, HA levels were comparable between Ctrl and $Slc35a3^{-/-}$ mice (Fig 6D). Because HA-synthases are localized in the plasma membrane [58], HA synthesis should not be affected by SLC35A3 deficiency.

SLC35A3 is reported to be involved in the biosynthesis of complex $N$-glycans. In $SLC35A3$-knockdown cells, the amounts of GlcNAc-branched tri- and tetra-antennary, but not mono- and di-antennary, $N$-glycans were decreased [15, 16]. Three studies have confirmed that SLC35A3 interacts with MGATs encoded by $MGAT4$ family member genes or $MGAT5$, which catalyze the branching to generate tri-and tetra-antennary $N$-glycans [15, 16, 54], suggesting that SLC35A3 regulates the biosynthesis of complex $N$-glycans by supplying the UDP-GlcNAc used as a substrate for the MGATs. Conversely, defects in the $N$-glycosylation pathway that result in hypoglycosylation lead to congenital disorders of $N$-linked glycosylation, which are clinically heterogeneous and include neurological symptoms such as epilepsy, intellectual

disability, myopathies, neuropathies and stroke-like episodes [59]. To date, the relationship between the defective biosynthesis of *N*-glycans, including the biosynthesis of highly branched *N*-glycans, and the neurological symptoms in AMRS and *N*-linked types of CDG has hardly been elucidated. Multiple analyses of the brain tissues in *Slc35a3*$^{-/-}$ mice, including structural elucidation of *N*-glycans, are currently in progress.

Because GlcNAc is one of the most common monosaccharide in the sugar chains of vertebral glycoproteins, it is possible that SLC35A3 is involved in the syntheses of various types of glycans. The pathological features caused by SLC35A3 deficiency differ between human and cattle. The *Slc35a3*$^{-/-}$ mice established in this study would be a useful model for investigating the *in vivo* role of SLC35A3 and the pathological mechanisms of SLC35A3-associated diseases, including CVM and AMRS.

## Supporting information

**S1 Fig. HPLC profiles of the chondroitinase digests of GAG-peptide preparations from the spine and limbs.** The 2-aminobenzamide (2AB)-derivatives of the yielded CS/DS and HA disaccharides after digestion with a mixture of chondroitinases ABC and AC-II, were separated using anion-exchange HPLC on an amine-bound silica PA-G column with a linear gradient of NaH$_2$PO$_4$ as indicated by the dashed line for the analysis of CS/DS and HA. HPLC profiles of the spine (A, B) and limb (C, D) samples of control (A, C) and *Slc35a3*$^{-/-}$ mice (B and D). The elution positions of 2AB-labeled CS/DS disaccharide standards are indicated by numbered arrows: 1, ΔHexUA-GalNAc; 2, ΔHexUA-GalNAc(6S); 3, ΔHexUA-GalNAc(4S); 4, ΔHexUA(2S)-GalNAc(6S); 5, ΔHexUA(2S)-GalNAc(4S); 6, ΔHexUA-GalNAc(4S,6S); 7, ΔHexUA(2S)-GalNAc(4S,6S). Asterisk indicates the ΔHexUA-GlcNAc derived from HA. Abbreviations: CS, chondroitin sulfate; DS, dermatan sulfate; HA, hyaluronan, ΔHexUA, 4,5-unsaturated hexuronic acid; GalNAc, *N*-acetyl-D-galactosamine; GlcNAc, *N*-acetyl-D-glucosamine; 2S, 2-*O*-sulfate; 4S, 4-*O*-sulfate; 6S, 6-*O*-sulfate.
(TIF)

**S2 Fig. HPLC profiles of the heparinase digests of GAG-peptide preparations from the spine and limbs.** The 2-aminobenzamide (2AB)-derivatives of the yielded HS disaccharides after digestion with a mixture of heparinase-I, heparitinase-II, and heparinase-III, were separated using anion-exchange HPLC on an amine-bound silica PA-G column with a linear gradient of NaH$_2$PO$_4$ as indicated by the dashed line for the analysis of HS. HPLC profiles of the spine (A, B) and limb (C, D) samples of control (A, C) and *Slc35a3*$^{-/-}$ mice (B, D). The elution positions of 2AB-labeled HS disaccharide standards are indicated by numbered arrows: 1, ΔHexUA-GlcNAc; 2, ΔHexUA-GlcNAc(6S); 3, ΔHexUA-GlcN(NS); 4, ΔHexUA-GlcN(NS,6S); 5, ΔHexUA(2S)-GlcN(NS); 6, ΔHexUA(2S)-GlcN(NS,6S). Abbreviations: HS, heparan sulfate; ΔHexUA, 4,5-unsaturated hexuronic acid; GlcNAc, *N*-acetyl-D-glucosamine; GlcN, D-glucosamine; 2S, 2-*O*-sulfate; 4S, 4-*O*-sulfate; 6S, 6-*O*-sulfate; NS, 2-*N*-sulfate.
(TIF)

**S3 Fig. HPLC profiles of the keratanase digests of GAG-peptide preparations from the spine.** The 2-aminobenzamide (2AB)-derivatives of the yielded KS disaccharides after digestion with a keratanase-II were separated using anion-exchange HPLC on an amine-bound silica PA-G column with a linear gradient of NaH$_2$PO$_4$ as indicated by the dashed line for analysis of KS. HLPC profiles of the spine samples of control (*A*) and *Slc35a3*$^{-/-}$ mice (*B*). The elution positions of 2AB-labeled KS disaccharide standards are indicated by numbered arrows: 1, Gal-GlcNAc(6S); 2, Gal(6S)-GlcNAc(6S). Abbreviation: KS, keratan sulfate; Gal, D-

galactose; GlcNAc, *N*-acetyl-D-glucosamine; 6S, 6-*O*-sulfate.
(TIF)

**S1 Raw images.**
(PDF)

**S1 Table. Disaccharide composition of chondroitin sulfate/dermatan sulfate (CS/DS) in the spine and limb.**
(TIF)

**S2 Table. Disaccharide composition of heparan sulfate (HS) in in the spine and limb.**
(TIF)

**S3 Table. Disaccharide composition of keratan sulfate (KS) disaccharides in the spine and limb.**
(TIF)

**S4 Table. Disaccharide level of hyaluronan (HA) disaccharides in in the spine and limb.**
(TIF)

## Acknowledgments

We thank Dr. Hirokazu Yagi for his helpful comments and discussions.

## Author Contributions

**Conceptualization:** Tatsuya Furuichi.

**Data curation:** Soichiro Saito, Shuji Mizumoto, Tsukasa Yonekura, Rina Yamashita, Kenta Nakano, Tadashi Okubo, Tadashi Okamura, Tatsuya Furuichi.

**Funding acquisition:** Shuji Mizumoto, Kenta Nakano, Shuhei Yamada, Tadashi Okamura, Tatsuya Furuichi.

**Investigation:** Soichiro Saito, Shuji Mizumoto, Tsukasa Yonekura, Rina Yamashita, Kenta Nakano, Tadashi Okubo, Shuhei Yamada, Tadashi Okamura, Tatsuya Furuichi.

**Methodology:** Shuji Mizumoto, Tadashi Okubo, Shuhei Yamada, Tadashi Okamura, Tatsuya Furuichi.

**Project administration:** Tatsuya Furuichi.

**Resources:** Tatsuya Furuichi.

**Supervision:** Tatsuya Furuichi.

**Validation:** Rina Yamashita.

**Visualization:** Soichiro Saito, Shuji Mizumoto, Tsukasa Yonekura, Tatsuya Furuichi.

**Writing – original draft:** Soichiro Saito, Shuji Mizumoto, Tsukasa Yonekura, Tatsuya Furuichi.

**Writing – review & editing:** Tatsuya Furuichi.

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
