## [Decision Letter · Decision Letter 0]

23 Jan 2023

PONE-D-22-34778Mice lacking nucleotide sugar transporter SLC35A3 exhibit lethal chondrodysplasia with vertebral anomalies and impaired glycosaminoglycan biosynthesisPLOS ONE

Dear Dr. Furuichi,

Thank you for submitting your manuscript to PLOS ONE. After careful consideration, we feel that it has merit but does not fully meet PLOS ONE’s publication criteria as it currently stands. Therefore, we invite you to submit a revised version of the manuscript that addresses the points raised during the review process.

The manuscript is well scheduled and performed. However, based on the Reviewers' comments some issues have arisen. Thus, the reviewers feel that it is well corroborated that SLC35A3 is involved in GAG synthesis but the differential effects of SLC35A3 on the different glycosylation types are lacking. The analysis of the N- and O- glycosylation status of the KO mice chondrocytes KO would strenghten the correlation. This needs to be responnded to and at least disccused. Furthermore how can the authors explain in their HPLC analysis (Supp figure 2) that the SLC35A3 deficiency affects more the peak 3, corresponding to ∆HexUA-GlcN(NS), than the peak 2 corresponding to ∆HexUA- GlcNAc(6S)?? Is it a bias due to the specificities of the enzymes used to cleave the GAGs chains? This needs to be disccussed. Furthermore, please elaborate on the technical querries and minor points raised by the reviewers.

We look forward to receiving your revised manuscript.

Kind regards,

Dragana Nikitovic, Ph.D

Academic Editor

PLOS ONE

2. To comply with PLOS ONE submissions requirements, in your Methods section, please provide additional information regarding the experiments involving animals and ensure you have included details on (1) methods of sacrifice for adult mice and mice embryos, (2) methods of anesthesia and/or analgesia, and (3) efforts to alleviate suffering.

3. As part of your revision, please complete and submit a copy of the Full ARRIVE 2.0 Guidelines checklist, a document that aims to improve experimental reporting and reproducibility of animal studies for purposes of post-publication data analysis and reproducibility: https://arriveguidelines.org/sites/arrive/files/documents/Author%20Checklist%20-%20Full.pdf (PDF). Please include your completed checklist as a Supporting Information file. Note that if your paper is accepted for publication, this checklist will be published as part of your article.

“We thank Dr. Hirokazu Yagi for his helpful comments and discussions. This study was supported by Grant-in-Aid from the Ito Foundation (T.F.). This study is also supported in part by a Grant-in-Aid for Science Research (C) from the Japan Society for the Promotion of Science (JSPS) (21K0593219 to T.F.; 19K07054 to S.M.; 21K065520 to S.Y.), Grant-in Aid from the Research Center for Pathogenesis of Intractable Diseases from the Research Institute of Meijo University (S.M. and S.Y.), and Grants-in-Aid for Research from the National Center for Global Health and Medicine (20A1019 and 20A2007D to T.O.(Okamura); 21A1018 to K.N.).”

“This study was supported by Grant-in-Aid from the Ito Foundation (T.F.). This study is also supported in part by a Grant-in-Aid for Science Research (C) from the Japan Society for the Promotion of Science (JSPS) (21K0593219 to T.F.; 19K07054 to S.M.; 21K065520 to S.Y.), Grant-in Aid from the Research Center for Pathogenesis of Intractable Diseases from the Research Institute of Meijo University (S.M. and S.Y.), and Grants-in-Aid for Research from the National Center for Global Health and Medicine (20A1019 and 20A2007D to T.O.(Okamura); 21A1018 to K.N.).The funders had no role in study design, data collection and analysis, decision to publish, or preparation of the manuscript.”

Additional Editor Comments:

The manuscript is well scheduled and performed. However, based on the Reviewers' comments some issues have arisen. Thus, the reviewers feel that it is well corrobortaed that SLC35A3 is involved in GAG synthesis but the differential effects of SLC35A3 on the different glycosylation types are lacking. The analysis of the N- and O- glycosylation status of the KO mice chondrocytes KO mice would strenghten the correlation. This needs to be responnded to and at least disccused. Furthermore how can the authors explain in their HPLC analysis (Supp figure 2) that the SLC35A3 deficiency affects more the peak 3 corresponding to ∆HexUA-GlcN(NS) than the peak 2 corresponding to ∆HexUA- GlcNAc(6S)?? Is it a bias due to the specificities of the enzymes used to cleave the GAGs chains? This needs to be disccussed. Furthermore, please elaborate on the technical querries and minor points raised by the reviewers.

Reviewers' comments:

Reviewer's Responses to Questions

**Comments to the Author**

1. Is the manuscript technically sound, and do the data support the conclusions?

Reviewer #1: Yes

Reviewer #2: Yes

2. Has the statistical analysis been performed appropriately and rigorously? 

Reviewer #1: Yes

Reviewer #2: Yes

3. Have the authors made all data underlying the findings in their manuscript fully available?

Reviewer #1: Yes

Reviewer #2: Yes

4. Is the manuscript presented in an intelligible fashion and written in standard English?

Reviewer #1: Yes

Reviewer #2: Yes

5. Review Comments to the Author

Reviewer #1: In this paper Saito et al generated a Slc35a3 knock-out mouse colony using the CRISPR/Cas9 genome editing system with the objective of elucidating the role of SLC35A3 in normal condition and in SLC35A3 related diseases.

Mutant mice were perinatal lethal with a severe chondrodysplasia phenotype including vertebral and growth plate anomalies caused by decreased GAG synthesis demonstrated by GAG disaccharide analysis.

The experiments are described in sufficient details and the approaches used to support the results are up to date.

MAJOR COMMENTS:

In the results, GAG synthesis in cartilage from mutant and wt mice is measured by quantitation of disaccharides from different GAG species in nmol/mg total protein. However in the results and also in the Material and Methods session it is not specified at which step of the experiment total proteins have been measured and the assay which has been used.

I’d suggest to mention that the RC, FC and HC zones in Fig 4 and 5 point to resting, proliferative and hypertrophic zones of the growth plate

MINOR COMMENTS

There are some inaccuracies that should be checked:

Page 8. Line 4 from bottom “lumber vertebrae”

Page 12 line 10 from top “embryos et E18.5” and line 15 from top “lumber vertebrae”

Page 14 line 16 from top “Sic35a3-/-“

Page 15 lines 6 and 7 from top “lumber vertebrae”

Page 18 line 11 from top “Slc35d-/-“

Reviewer #2: The paper of Soichiro Saito and collaborators investigated the relationship between SLC35A3 and GAG synthesis by generating a KO SLC35A3 mice. They nicely showed that mice exhibited lethal chondrodysplasia with vertebral anomalies and impaired glycosaminoglycan biosynthesis. This is an interesting paper bringing more evidences that SLC35A3 is involved in GAG synthesis. The first part of the paper is very strong and well done with the generation and the KO SLC35A3 and its phenotyping. The second part on the GAG analysis is more difficult to follow and constitutes the weakest part of the paper. I understand the difficulty of these analysis but as it is, the paper stays descriptive. The mouse recapitulates the SLC35A3 human phenotype with major skeletal abnormalities due to GAG defects. This is known that SLC35A3 is involved in GAG synthesis (see my minor point). I also have the feeling that we miss the fundamental study on trying to explain the differential effects of a lack of SLC35A3 on the different glycosylation types. It would have brought a lot if the authors could have analyzed the N- and O- glycosylation status of the chondrocytes KO mice. This is my major criticism.

Major point:

Coming back to the GAGs analysis by HPLC after enzymatic digestions, how can the authors explain in their analysis (Supp figure 2) that a deficiency affects more the pick 3 corresponding to ∆HexUA-GlcN(NS) than the pick 2 corresponding to ∆HexUA- GlcNAc(6S)?? Is it a bias due to the specificities of the enzymes used to cleave the GAGs chains? This can be annoying.

I fully agree with a general GAGs defect from the experiments but the differential quantification defect on HS/ KS/ DS is much more difficult to assess from the experiments. This “strong” GalNAc defect is also interesting but stays descriptive. This is where O-GalNAc glycosylation analysis would be very interesting in confirming this defect.

Minor point:

The authors say that “the relationship between the function of SLC35A3 and GAG biosynthesis is largely unknown”. I do not agree with this sentence. The link has been evidenced in Haouari’s paper on serum bikunin isoforms in congenital disorders of glycosylation and linkeropathies. The clear defect observed on bikunin in SLC35A3 patient strongly demonstrates that GAGs are affected.

6. PLOS authors have the option to publish the peer review history of their article (what does this mean?). If published, this will include your full peer review and any attached files.

Reviewer #1: No

Reviewer #2: No

---

## [Author Response · Author response to Decision Letter 0]

24 Feb 2023

<To Reviewer #1>

Comment: In the results, GAG synthesis in cartilage from mutant and wt mice is measured by quantitation of disaccharides from different GAG species in nmol/mg total protein. However in the results and also in the Material and Methods session it is not specified at which step of the experiment total proteins have been measured and the assay which has been used.

Response: We added the following sentence in page 9: “Total amount of proteins in the sonicates was determined by micro BCA protein assay kit (Thermo Fisher Scientific)”.

Comment: I’d mention that the RC, FC and HC zones in Fig 4 and 5 point to resting, proliferative and hypertrophic zones of the growth plate.

Response: We described the RC, FC and HC zones in the Figure legends as following in page 14: High magnification images of the round chondrocyte (RC), flat chondrocyte (FC), and hypertrophic chondrocyte (HC) zones in the growth plates, RC (Fig 4 legend), round chondrocyte zone; FC, flat chondrocyte zone (Fig 5 legend).

Comment: There are some inaccuracies that should be checked:

Page 8. Line 4 from bottom “lumber vertebrae”

Page 12 line 10 from top “embryos et E18.5” and line 15 from top “lumber vertebrae”

Page 14 line 16 from top “Sic35a3-/-“

Page 15 lines 6 and 7 from top “lumber vertebrae”

Response: We corrected these errors in page 11, 12 and 14.

Comment: There are some inaccuracies that should be checked:

Page 18 line 11 from top “Slc35d-/-“

Response: The original is correct (page 18).

<To Reviewer #2>

Comment: Coming back to the GAGs analysis by HPLC after enzymatic digestions, how can the authors explain in their analysis (Supp figure 2) that a deficiency affects more the pick 3 corresponding to ∆HexUA-GlcN(NS) than the pick 2 corresponding to ∆HexUA- GlcNAc(6S)?? Is it a bias due to the specificities of the enzymes used to cleave the GAGs chains? This can be annoying.

Response: We thank reviewer#2 for the constructive comment. The greater reduction of ∆HexUA-GlcN(NS) than ∆HexUA-GlcNAc(6S) in Slc35a3–/– samples does not seem to be caused by the bias of HS-degrading enzymes, because the mixture of heparinases -I, -II, and -III can act equally on the HexUA-GlcN(NS) and HexUA-GlcNAc(6S)-containing sequences. The cause is unknown, but we would like to propose the following as one possibility. SLC35A3 is known to form huge complexes involved mannoside acetylglucosaminyltransferases (Mgats) and affect Mgat activities. Therefore, SLC35A3 may also form complexes with the HS-polymerase, a hetero-complex of EXT1 and EXT2, and affect their activities. The sugar backbone of HS, -GlcUA-GlcNAc-, is formed by EXT1/EXT2, and then N-deacetylase/N-sulfotransferase-1 (NDST1) catalyzes N-deacetylation and N-sulfation of GlcNAc residue in HS, resulting in the formation of HexUA-GlcN(NS)-containing sequence. EXT2 is reported to bind with NDST1 and regulate NDAT1 activity to form HexUA-GlcN(NS) (Presto, et al., Proc Natl Acad Sci USA 105: 4751-6, 2008). Therefore, SLC35A3 deficiency may cause not only depletion of UDP-GlcNAc but also the impairment of collaborative interaction of EXT1/EXT2 with NDST1 in the Golgi apparatus, which seems to lead to a decrease in conversion from HexUA-GlcNAc to HexUA-GlcN(NS). Consequently, the amount of HexUA-GlcN(NS) may be drastically decreased in Slc35a3–/– mice. In contrast, HS 6-O-sulfotransferase (HS6ST) catalyzes the transfer of sulfate to the C-6 position of GlcNAc residue in a non-cooperative way with EXT1/EXT2. Therefore, the conversion from HexUA-GlcNAc to HexUA-GlcNAc(6S) may not be severely affected in in Slc35a3–/– mice.

Currently, the investigation to identify multiprotein complexes involving SLC35A3 in cartilage is in progress, probably leading to the elucidation of this unclear mechanism.

Comment: I fully agree with a general GAGs defect from the experiments but the differential quantification defect on HS/ KS/ DS is much more difficult to assess from the experiments. This “strong” GalNAc defect is also interesting but stays descriptive. This is where O-GalNAc glycosylation analysis would be very interesting in confirming this defect.

Response: We are also interested in the O-linked fucose containing structures on Notch receptors in Slc35a3–/– mice. However, the endogenous O-linked fucose structures on Notch receptors could not be determined in mice because the in vitro glycosyltransferase assays using radiolabelling glycan technique and overexpression of the glycosyltransferases in CHO cell line that cannot synthesis complex or hybrid-type-N-glycans are essential to determine it (Moloney DJ et al, Nature 406: 369–, 2000). This paper focuses on analysis of the skeletal tissues (cartilage and spine) in Slc35a3–/– mice. The amounts of N-glycan and O-glycan in skeletal tissues are very small; therefore, few papers reported the roles of N-glycans and O-glycans in skeletal tissues. SLC35A3 is reported to regulate the branching of N-glycans, and AMRS caused by SLC35A3 mutations and congenital disorders of N-linked glycosylation are characterized by mental retardation and epilepsy. Therefore, we are interested in the interactions of SLC35A3 functions and N-glycan metabolisms in brain and have initiated the studies to examine them. It will take some time to obtain the results, and so we hope to report the results in our second paper. We have provided this information in page 19-20. We would like to take the interactions between SLC35A3 and O-glycan as the future consideration.

Comment: 2. The authors say that “the relationship between the function of SLC35A3 and GAG biosynthesis is largely unknown”. I do not agree with this sentence. The link has been evidenced in Haouari’s paper on serum bikunin isoforms in congenital disorders of glycosylation and linkeropathies. The clear defect observed on bikunin in SLC35A3 patient strongly demonstrates that GAGs are affected

Response: We accept this comment. " is largely unknown " was changed to " is not fully clarified" in page 5.

<To Editor>

Comment: To comply with PLOS ONE submissions requirements, in your Methods section, please provide additional information regarding the experiments involving animals and ensure you have included details on (1) methods of sacrifice for adult mice and mice embryos, (2) methods of anesthesia and/or analgesia, and (3) efforts to alleviate suffering.

Response: We added the following sentence in page 5: “Mice were sacrificed by an overdose of pentobarbital or cervical dislocation under isoflurane anesthesia. Mouse embryos were sacrificed by decapitation.”.

Comment: Please remove any funding-related text from the manuscript and let us know how you would like to update your Funding Statement. Currently, your Funding Statement reads as follows:

Response: We deleted the following in page 20:“This study was supported by Grant-in-Aid from the Ito Foundation (T.F.). This study is also supported in part by a Grant-in-Aid for Science Research (C) from the Japan Society for the Promotion of Science (JSPS) (21K0593219 to T.F.; 19K07054 to S.M.; 21K065520 to S.Y.), Grant-in Aid from the Research Center for Pathogenesis of Intractable Diseases from the Research Institute of Meijo University (S.M. and S.Y.), and Grants-in-Aid for Research from the National Center for Global Health and Medicine (20A1019 and 20A2007D to T.O.(Okamura); 21A1018 to K.N.).The funders had no role in study design, data collection and analysis, decision to publish, or preparation of the manuscript.”

Comment: In your cover letter, please note whether your blot/gel image data are in Supporting Information or posted at a public data repository, provide the repository URL if relevant, and provide specific details as to which raw blot/gel images,

Response: We provided the original image for Fig 1B in Supporting Information.

---

## [Decision Letter · Decision Letter 1]

28 Mar 2023

Mice lacking nucleotide sugar transporter SLC35A3 exhibit lethal chondrodysplasia with vertebral anomalies and impaired glycosaminoglycan biosynthesis

PONE-D-22-34778R1

Dear Dr. Furuichi,

We’re pleased to inform you that your manuscript has been judged scientifically suitable for publication and will be formally accepted for publication once it meets all outstanding technical requirements.

Kind regards,

Dragana Nikitovic, Ph.D

Academic Editor

PLOS ONE

Additional Editor Comments (optional):

Reviewers' comments:

Reviewer's Responses to Questions

**Comments to the Author**

1. If the authors have adequately addressed your comments raised in a previous round of review and you feel that this manuscript is now acceptable for publication, you may indicate that here to bypass the “Comments to the Author” section, enter your conflict of interest statement in the “Confidential to Editor” section, and submit your "Accept" recommendation.

Reviewer #1: All comments have been addressed

Reviewer #2: All comments have been addressed

2. Is the manuscript technically sound, and do the data support the conclusions?

Reviewer #1: Yes

Reviewer #2: Yes

3. Has the statistical analysis been performed appropriately and rigorously? 

Reviewer #1: Yes

Reviewer #2: Yes

4. Have the authors made all data underlying the findings in their manuscript fully available?

Reviewer #1: Yes

Reviewer #2: Yes

5. Is the manuscript presented in an intelligible fashion and written in standard English?

Reviewer #1: Yes

Reviewer #2: Yes

6. Review Comments to the Author

Reviewer #1: (No Response)

Reviewer #2: (No Response)

7. PLOS authors have the option to publish the peer review history of their article (what does this mean?). If published, this will include your full peer review and any attached files.

Reviewer #1: No

Reviewer #2: No

---

## [Editor Report · Acceptance letter]

5 Apr 2023

PONE-D-22-34778R1 

Mice lacking nucleotide sugar transporter SLC35A3 exhibit lethal chondrodysplasia with vertebral anomalies and impaired glycosaminoglycan biosynthesis 

Dear Dr. Furuichi:

I'm pleased to inform you that your manuscript has been deemed suitable for publication in PLOS ONE. Congratulations! Your manuscript is now with our production department. 

Kind regards, 

on behalf of

Dr. Dragana Nikitovic 

Academic Editor

PLOS ONE